# Factors Associated with Physical Activity in a Diverse Older Population

**DOI:** 10.3390/geriatrics7050111

**Published:** 2022-10-04

**Authors:** Ruth M. Tappen, David Newman, Sareen S. Gropper, Cassandre Horne, Edgar R. Vieira

**Affiliations:** 1Christine E. Lynn College of Nursing, Florida Atlantic University, 777 Glades Road NU-84, Boca Raton, FL 33431, USA; 2Nicole Wertheim College of Nursing & Health Sciences, Florida International University, 11200 SW 8th St., AHC3-430, Miami, FL 33199, USA

**Keywords:** physical activity, older adults, determinants, pain, social network, depression, elastic net, multicollinearity

## Abstract

Physical activity is important for healthy aging, but few older adults achieve the goal of 150 min per week of moderate activity. The purpose of this study was to employ a robust statistical approach in the analysis of the factors related to physical activity in a diverse sample of older adults. A secondary analysis of factors associated with calculated MET-h/week was conducted in a sample of 601 African Americans, Afro-Caribbeans, European Americans, and Hispanic Americans age 59 to 96 living independently in the community. Age, education, social network, pain, and depression were the five variables that accounted for a statistically significant proportion of unique variance in the model. The strongest correlation to total MET-h/week was with depression. Directionality of the relationship between these variables and physical activity is complex: while pain and depression can reduce physical activity, activity may also help to reduce pain and depression. Additionally, of note is that many of these factors may be modified, calling for the design and testing of individual, group, and community level interventions to increase physical activity in the older population.

## 1. Introduction

Physical activity is important for healthy aging [1,2]. Regular physical activity helps prevent functional decline, frailty, falls and chronic conditions such as diabetes and cardiovascular disease [3]. It contributes to quality of life and reduced depression. Multiple factors affect physical activity levels among older adults and differences among various racial and ethnic groups have been reported [4,5]. Implicated factors include demographics, socioeconomics, social determinants of health, environmental conditions, health status and physical function [6]. Prior studies evaluating factors associated with physical activity among older adults have reported conflicting results and did not account for collinearity among variables.

Despite the known health benefits associated with regular physical activity, older adults seldom meet physical activity guidelines. Thus, it is not surprising that research studies, both qualitative and quantitative, undertaking to identify barriers and motivators influencing activity participation by older adults are abundant in the scientific literature. A systematic review conducted by Notthoff, Reisch and Gerstorf [7] in 2017, for example, screened over 21,000 peer-reviewed studies published between January 1995 and September 2016 with the objective of identifying individual characteristics consistently associated with high physical activity levels in older adults. These researchers identified two psychological factors (motivation and self-efficacy) as well as two demographic variables (gender and education) that were most consistently linked with higher physical activity. This review, however, did not report information on the ethnic group membership of the sample populations in the included studies. Among more recently reported research, there are limited studies examining factors associated with physical activity in ethnically diverse groups of older adults.

Gomes [8] studied physical inactivity among 19,298 older adults across 12 European countries. The authors found that the prevalence of inactivity was 12.5%, varying from 5% in Sweden to 29% in Portugal. The primary factors associated with inactivity were older age, depression, physical impairments, low meaning in life, low social support, and memory loss. However, the authors did not investigate differences among ethnic groups. A recently published study by AlFaris [9] evaluated physical activity levels and factors associated with activity and inactivity in a multi-ethnic population of 1800 middle-aged men in Saudi Arabia. The authors found that the lowest prevalence of physical activity was 16% in those from the Philippines and the highest was 58% in those from Saudi Arabia, suggesting immigrants may be more inactive. Other factors associated with inactivity were family living, educational level, income, and body mass index (BMI). However, these authors did not investigate the effects of different factors among the ethnic groups, and they did not study older adults. Another study evaluated if social isolation, ethnicity, and gender were associated with inactivity in 3298 middle and older age adults living in Northern Manhattan [10]. The prevalence of inactivity was 40.5%. Inactivity was more prevalent among Hispanics (OR: 2.18), in women (OR: 1.33), in those on Medicaid or uninsured (OR: 1.2), and in those who had less than 3 friends (OR: 1.41). This study focused only on social factors controlling for co-morbidities. Additional evaluation of the factors that affect physical activity levels in older adults is still needed.

Understanding what factors affect physical activity levels among older adults from different ethnic groups can inform the development of interventions to encourage and support regular physical activity among older adults. Therefore, the objectives of this study were to identify factors associated with different amounts of physical activity among a large, diverse sample of older adults living independently in the community using robust statistical methods that account for multicollinearity among many of these factors.

## 2. Materials and Methods

### 2.1. Design

A secondary analysis of the physical and psychosocial correlates of physical activity was conducted using data from the Healthy Aging Research Initiative (HARI) an in-depth survey of the health and well-being of multiethnic community dwelling older adults in South Florida. Participants were drawn from three metropolitan areas encompassing Miami-Dade, Broward, and Palm Beach counties [11,12]. The population of these areas is primarily African American, Afro-Caribbean, European American, and Hispanic American, with a substantial proportion of individuals who were first generation Americans, i.e., born outside the United States. Three to four separate interview sessions for assessment of physical and psychosocial status and health-related behaviors including physical activity were conducted by bilingual English-Spanish and English-Creole members of the research team.

### 2.2. Sample

The HARI database contains information on 698 participants, of which 601 had complete data on their physical activity level and the related physical and psychosocial variables of interest. This included 115 African Americans, 142 Afro-Caribbeans, 217 European Americans, and 127 Hispanic Americans ranging in age from 59 (approaching 60) to 96. All were living independently in the community, able to walk at least 6 m independently, or with the aid of a cane or walker, and had a Mini-Mental State Examination score ≥ 23 [13]. Interviews were conducted in English, Spanish, and Haitian Creole. The study was approved by the University Committee for the Protection of Human Subjects (IRB). Written consent was obtained from every participant after an explanation of the study was provided.

### 2.3. Data Collection

Sociodemographic variables included age, sex, years of education, ethnic group membership, years living in the United States, and receipt of Medicaid based upon income level qualifications. Cognition was measured using the Mini-Mental State Exam [13]. Psychosocial variables included social engagement, social network, resilience, personality, anxiety, depression, spirituality, and the SF-36 mental health summary score [14,15,16,17,18,19,20,21,22,23]. Physical measures included pain, body mass index, body consciousness, functional ability, and self-rating of health [24,25,26,27,28]. Behavioral variables included adherence to prescribed medications and self-reported physical activity levels [24,29]. The specific measures used and their sources are presented in Table 1.

### 2.4. Calculation of Physical Activity Index

Activity was calculated from three variables derived from Health and Retirement Study [24] items measuring the frequency within a month that participants engaged in low, moderate and high intensity activity. Within each of these categories, frequency choices were daily, several times a week, once a week, several times a month, at least once a month or not at all in the last month. Based on suggestions from Smith et al. (2017) [30], the frequency of activities at each level was tabulated. The METs (Metabolic Equivalent Table) per level were calculated on the basis of reported frequency of activity per level: low physical activity = 1.6 METS, moderate physical activity = 3.0 METs and high physical activity = 6 METs for 30 min of activity. Then, the total METs per hour per week (MET-h/week) were calculated by dividing the total METs by 60 [31]. For this study the total MET h per week ranged from a high of 10.6 MET-h/week to a low of 0.75 MET-h/week with an average of 4.4 MET-h/week. This is lower than the recommendation from the American Heart Association of a minimum 150 min of physical activity at a moderate level. At a moderate level (3 METs) this is equal to 7.5 MET-h/week [32]. Participants were also grouped by weekly activity level (low, medium, or high) calculated in accordance with CDC and ACSM guidelines with less than an average of 3 MET-h/week as low activity, 3 to 6 MET-h/week as moderate activity and more than 6 as high activity [33,34].

### 2.5. Data Analysis

Sociodemographic, cognitive, psychosocial, physical, and behavioral variables were selected for analysis from the HARI database with the goal of building the most parsimonious and best fitting model to predict the older adult participant’s reported frequency and intensity of physical activity. To build this model, elastic net (EN) regression was used. EN is a regularization technique that combines the strengths of both ridge regression and least absolute shrinkage and selection operator (LASSO) regression models. This is especially important in cases of high multicollinearity where ordinary least squares (OLS) models may yield unstable and unreliable parameter estimates. Additionally, EN outperforms OLS when you have a high number of predictors compared to the number of participants. This occurs because OLS models also have higher variability as the number of predictors increases [35,36].

With ridge regression, the traditional OLS model is adjusted to both the sum of squared residuals as well as to the penalty size related to the number of predictors made to the parameter estimates, in order to shrink them towards zero. For ridge regression the λ parameter represents the regularization penalty. LASSO, least absolute shrinkage and selection operator, is quite similar conceptually to ridge regression. It also adds a penalty for non-zero coefficients, but unlike ridge regression which penalizes the sum of squared coefficients (the L2 penalty), LASSO penalizes the sum of their absolute values (L1 penalty). As a result, with higher values of λ many coefficients are exactly zeroed under LASSO, which is never the case in ridge regression. Elastic net (EN) first emerged as a result of critique of LASSO, whose variable selection can be too dependent on the data and thus unstable. The solution is to combine the penalties of ridge regression and LASSO to get the best of both worlds [37]. Cross-validation (CV) was used to identify the optimal degree of penalty for L1 and L2. This CV used a 0.632 bootstrapped method to draw 100 new samples with a 63.2% similarity to our original sample. The models selected used the minimum mean-squared error (MSE) [38].

## 3. Results

### 3.1. Sample Characteristics

The age of the participants in this study ranged from 59 (one participant approaching the 60th birthday) to 96 years old with an average age of 74.3 years. All of the participants had lived in the United States for at least one year with an average of 55.5 years. On a pain scale of 0 to 10, the average reported pain was low (2.9). However, there were 94 participants who reported their average pain at or above an eight out of ten. BMI ranged from 16 to 48 with an average of 29. The average activity index was 4.4 METS-h/week.

As can be seen in Table 2, there were statistical differences across ethnic groups in age (*p* < 0.001), years of education (*p* < 0.001), years in the US (*p* < 0.001), cognition (*p* < 0.001), functional activity (*p* < 0.001), and calculated activity index (*p* < 0.002). European Americans reported statistically significantly higher average physical activity levels (4.8) than African Americans (4.1), Afro-Caribbeans (4.7) or Hispanics (3.6) [F = (3598) = 5.10, *p* = 0.002, η^2^ = 0.01].

### 3.2. Differences by Activity Level

The next step in building our predictive model was to assess differences in physical activity level across the demographic, cognitive, psychosocial, physical and behavioral variables that emerged from prior qualitative analysis of a smaller sample of diverse older adults [12] and from the literature. As can be seen in Table 3 there were significant differences in 13 of the 17 predictive variables across the activity levels of low, medium, and high as outlined by the AHA guidelines [32] (Low < 3 Mets, Medium 3-6 Mets, High over 6 Mets). These variables include age (*p* = 0.001), years of education (*p* = 0.002), social engagement (*p* < 0.001), social network (*p* < 0.001), resilience (*p* < 0.001), depression (*p* < 0.001), anxiety (*p* < 0.003), SF-36 mental health summary score (*p* < 0.024), cognition (*p* = 0.002), pain (*p* = 0.004), BMI (*p* < 0.001) self-health rating (*p* = 0.011), and functional activity (*p* < 0.001). Participants who reported lower physical activity tended to be older, have less years of education, and reported lower social engagement, networking, resilience, mental health, self-health rating, and higher levels of depression, anxiety, pain, and BMI compared to the moderate to high physical activity groups.

After identifying the potential differences by ethnic group and activity level, these variables in combination were included in a predictive model to account for the variability reported in the activity index as measured by MET-h/per week. Elastic net regression (EN) utilized cross validation to obtain the best, most parsimonious and stable, model. A repeated tenfold cross-validation with a 0.1 standard error was used to identify the most appropriate shrinkage parameter (λ) as the final selection criteria for the most appropriate model. The EN model suggested the best fitting λ = 1 and LASSO penalty = 0.180 and retained eleven variables that accounted for 22.2% of the variability in predicting the MET-h/week. Six of these accounted for unique variance while the other five did not and were removed from the model.

Table 4 compares the OLS results from the 25 variables to those of the EN Regression. It can be seen in the EN Model that 19 of the original 25 variables standardized coefficients (*β*) set to zero essentially removing them from the model. As mentioned earlier, this is the variable selection technique of importance. The six variables retained through this regularization process were age, social network, depression, cognition, pain and BMI which accounted for a statistically significant proportion of unique variance (R = 0.470, R^2^ = 0.221 and SE = 0.035). Of the six variables identified by the EN regression, five variables accounted for a statistically significant proportion of unique variance (*p* = 0.05), age, years of education, social network, depression, and pain.

A Pearson correlation was conducted on the five variables that accounted for a statistically significant proportion of unique variance in the EN. There were statistically significant correlations with total MET-h/week (see Table 5). The strongest correlation was for depression (r = −0.186, *p* < 0.001). As can be seen in this table, even though these variables were retained by the EN there are still significant relationships between many of the predictors such as depression and education (r = −0.189, *p* < 0.001), depression and social network (r = −0.198, *p* < 0.001), and pain and depression (r = −0.227, *p* < 0.001).

These five variables were then refitted using OLS regression to obtain the unpenalized standardized and unstandardized coefficients. As can be seen in Table 6 even though all of these variables were retained only depression (B = −1.53, *p* < 0.001) accounted for a significant proportion of unique variance in predicting the MET-h/week (*R*^2^ = 0.165). Depression accounted for the largest proportion of unique variance in the model.

## 4. Discussion

The purpose of this secondary analysis was to examine factors associated with physical activity in a diverse older population. Unique to this study was the use of rigorous analytical techniques to build the most parsimonious, best fitting model predicting older adults’ levels of physical activity. This is particularly important in instances of a large number of variables with high multicollinearity. This study used elastic net regression analysis combining the penalties for both ridge regression and LASSO to investigate the underlying factors that predict physical activity in older adults. For example, instead of just studying ethnic differences in physical activity, we went further and identified the key significant predictors, many of which were highly correlated with ethnicity.

Additionally, when controlling for variability accounted for by these underlying factors, the unique variance accounted for by ethnicity alone becomes nonsignificant. The elastic net regression technique is less affected by multicollinearity between these underlying components and ethnicity and therefore allowed us to get a clearer view of these critical underlying components. While a number of published studies have addressed the question of factors influencing older adults’ physical activity levels, none have employed the large range of instruments/tools, used this robust statistical analysis, and included older adults from multiple ethnic groups as done in this study.

Other published studies conducted in older adults have targeted population groups in countries such as Ireland [39], Jamaica [40], Scotland [41], India [42], Australia [43], and Malaysia [44] among others. There is a scarcity of research directed at studying differences in factors associated with physical activity within groups (including men and women) based on race or ethnicity. The present study helped to fill this gap, especially with the inclusion of Afro-Caribbeans, a group that has rarely been examined independent of African Americans. Interestingly, our study found that while ethnic group membership was a factor based on a one-way analysis of variance (ANOVA), more rigorous analysis across the many potential predictor variables eliminated ethnic group membership and identified five factors-age, fewer years of education, smaller social network, higher depression, and higher pain—as major contributors to lower participation in physical activity among ethnically diverse older adults. Social network, depression, and pain can be ameliorated by physical activity. They are modifiable factors that can be addressed by physical activity itself. The effects of limited years of education are also modifiable, but this is more difficult, and cannot be addressed by physical activity itself.

Some of the findings from this study, including the impact of age, depression, and social network (which also included social support), have been confirmed in the literature. McKee, Kearney and Kenny [39], for example, investigated factors associated with participation in physical activity among 3499 adults aged 65 years and older living in Ireland. The studied variables included socio-demographics, social connectedness, physical environment and several physical and mental health related factors. Using bivariate analysis followed by multivariate analysis, the final model identified 13 variables that remained significantly associated with lower physical activity. These variables were more time spent sitting, being female, being older, having a higher depression score, having a lower quality of life, lower grip test, reduced activities of daily living, higher BMI, higher anxiety score, not employed, less social connectedness, not living in a detached house, and having lower cognitive function score.

Similar to the findings from the present study, other studies investigating older men and/or women have identified education, age, social support, and depression as factors contributing to physical inactivity. For example, age, education and social support along with current health and employment status, were identified as being associated with physical activity among older men residing in Jamaica [40]. Higher levels of education, among other factors, have also been linked with vigorous physical activity in women aged 60 years and older, residing in Spain [45], while mild levels of exercise were associated with frequency of social relationships and greater satisfaction in a group of 257 women age 61–93 years residing in Spain [46]. Older age, depressive symptoms and some environmental factors were also found in bivariate analysis to be negatively associated with physical activity among 102 older African American and European American women [43]. Depression has been linked with physical inactivity in older adults in multiple studies [8,39,47]. Pain, especially musculoskeletal pain, was also found to be highly associated with sedentary behavior among older adults from six low- and middle-income countries [48]. Other variables that were not found to be linked were education, social support, marital status, self-efficacy, perceived stress, and perceived neighborhood safety.

Some studies have also reported pain as a factor that influenced physical activity. Among a group of 409 older adults living independently in Scotland, using logistic regression and then regression modelling, Crombie and colleagues [41] showed that the following factors exerted significant independent effects on physical activity: joint pain; not belonging to a group; lack of interest, fitness, energy, and access to a car; shortness of breath, dislike of going out alone, disbelief that exercise affects lifespan, and doubt that meeting new people is beneficial. Depression, age, and years of education were not examined in this study.

In one of the few studies aimed at identifying factors associated with physical activity that included multiple ethnic groups, advanced age and fewer years of education (as in the present study) were linked with inactivity in a group of 2912 middle- and older-aged (40 years and older) American Indian, African American, Hispanic and European American women [49]. This study, using multivariate regression and logistic regression analyses, showed that physical inactivity was significantly associated with American Indian ethnicity, lack of energy, neighborhood environment, and infrequent observation of others exercising. Other identified factors associated with less physical activity included caregiving duties, poor health, being too tired, and inadequate support. Self-consciousness was related to increased activity.

### 4.1. Implications for Practice

In the final model, a set of five predictors of activity were retained: age, years of education, social activity, depression, and pain. Interestingly, many of these underlying factors are modifiable. The implications of these predictors are discussed in the following paragraphs with an emphasis on those that can be addressed through individual and/or community level intervention.

The findings from the current study suggest development of interventions and programs aimed at increasing older adults’ participation in physical activity. While age is not modifiable, depressive symptoms and pain can often be improved with physical activity and social networks can be created or expanded. Tailored activity sessions can be developed and provided to encourage inactive older adults to begin and/or increase physical activity. Education may be key both to helping older adults with depressive symptoms understand that physical activity can help reduce their symptoms and in helping them identify the types of activity that they may find enjoyable. Marques et al. [50] reported that participation by both men and women in moderate or vigorous physical activity as little as once a week was associated with lower depression symptom scores. Similar findings have been reported by Mumba and others [51], while other studies evaluated the effectiveness of physical exercise as an alternative to medication in the treatment of depression in older adults [52].

The present study, like others, found that pain was also associated with less time spent being physically active. What is not clear is whether older adults understand that sedentary lifestyles can promote and/or worsen some types of pain [53] and physical activity can help to reduce pain [54,55] or whether this knowledge alone is enough to motivate them to become more active. A study by Tappen and colleagues [12] reported on interviews of a group of ethnically diverse older adults that physical activity sustained over time was often done because of a personal goal or purpose. Primary care providers and other health professionals are potentially important in providing encouragement, identifying personally meaningful goals and information on physical activity for pain relief and reducing depressive symptoms [56,57].

Social support and connectedness, identified as a factor influencing participation in physical activity in this study and other studies [12,39,41,48] is also modifiable for those willing to engage socially [58]. While researchers in Australia reported from a group of over 12,000 adults that social support was a weak predictor of physical activity, among a subgroup of 927 people with conditions that caused recurring or chronic pain, social support had a significant indirect effect of helping to attenuate the pain. Older adults participating in group-based physical activity have been shown to exhibit decreased depressive symptoms [59,60]. Further, a systematic review examining social support and physical activity in older adults reported that those with more social support (especially from family members) were more likely to participate in leisure time physical activity [61]. Encouraging older adults to have a family member(s) or friend(s) join them in beginning a physical activity may represent a means of facilitating initial participation. Encouraging participation in group exercise classes may also be helpful for those in need of social support given that friendships often develop within group exercise classes. In these instances, social support often becomes directly linked with participation in the class and can provide a sense of belonging which has been linked with assisting older adults in maintaining adherence to group-based physical activity classes [62]. Mutual reinforcement within and across these activities can contribute to the health and quality of life of older adults.

A complex group of interacting factors that influences participation in physical activity among older adults clearly exists. While these factors may be challenging to overcome, involvement in physical activity can provide enormous benefits including long-term reduction in pain and symptoms of depression and improved social connectiveness among others. Partnerships among local senior centers, low income housing developments, places of worship, YMCAs, and healthcare providers are crucial in the development of tailored multi-faceted programs for physically inactive older adults especially those experiencing pain and/or depression. These programs can provide health-related education pertinent to the identified medical issues (e.g., pain, depression) and assist participants in meeting other participants and in developing specific physical activity-related goals which are known to be associated with sustained involvement. [63] Program locations need to be in safe areas and easily accessible. Moreover, instructors teaching the physical activities/exercises need to ensure that participants are demonstrating correct form to prevent further injury and/or to address any muscle and/or bone-related medical conditions.

### 4.2. Limitations

There are several important limitations to this study. The influence of motivation and self-efficacy could not be evaluated as these variables were not in the HARI database. Similarly, the effect of the environmental context on engagement in physical activity was not evaluated. This includes the walkability of the neighborhood, access to safe, user friendly transportation, free or low-cost exercise programs, and the existence of safety concerns that impede neighborhood activity and interaction. Further, the directionality of several of the significant predictors cannot be determined in a cross-sectional study. One example is the interaction of depression and physical activity: those who are depressed are less likely to engage in physical activity but activity has also been cited as an efficient nonpharmacological intervention for depression. The interrelationship between pain and physical activity is another portion of this cycle of pain, depression, and inactivity that affects many older adults [53].

### 4.3. Implications for Future Research

Both the results of this study and the limitations of this work suggest direction for future research. The bidirectionality of the interactions between some of the variables of interest and need to know, for example, in what instances pain or depression precedes/leads to inactivity versus resulting from inactivity, calls for examination of longitudinal data to understand these relationships. The high degree of interaction among many of the factors affecting physical activity suggest that future studies take a multifactorial, holistic approach encompassing all of the known and even suggested predictors to produce a comprehensive model for physical activity in older adults.

Four of the five significant predictors of physical activity in the older adults studied are at least partially modifiable, calling for the testing of interventions addressing these factors’ potential effect on physical activity and inactivity. A few suggestions include education regarding the effect of activity on common sources of pain such as arthritis or back pain; encouraging providers to write a “prescription” for a daily walk or workout for those with depression, community outreach to isolated older adults, improving the walkability of neighborhoods, repairing sidewalks, adding trails, and making these areas safe to walk and work out.

## 5. Conclusions

Age, education, social network, pain, and depression were the five factors that accounted for a statistically significant proportion of unique variance in physical activity in this diverse, community dwelling older population. The bidirectionality of their effect is noted as the presence of pain or depression is associated with lower activity, but physical activity has been shown to reduce pain and depression as well. Longitudinal studies would help understand these relationships. Furthermore, most of these factors are also modifiable, calling for greater efforts to design, test, and implement individual, group, and community level interventions that support physical exercise and encourage older adults to increase their activity.

## Figures and Tables

**Table 1 geriatrics-07-00111-t001:** Measures Included in Predictive Model.

Domain	Measurement	Description
Sociodemographic Information	Age	In Years
	Gender	Male-Female
	Education	Years in school
	Residency in U.S.	In years
	Income	Receiving Medicaid
Cognition	Mini-Mental State Exam	Global cognitive screening [13]
Psychosocial Parameters	Social Engagement	Productive and leisure activities [14]
	Social Network	Lubben Social Network Scale-6 (LSNS-6) [15,16]
	Spirituality	Spirituality Perspective Scale (SPS) [17]
	Resilience	Connor-Davidson Resilience Scale (CD-RISC) [18]
	Depression	Center for Epidemiologic Studies Depression Scale (CES-D) [19]
	Anxiety	Self-Evaluation Questionnaire [20]
	Personality	Big Five Inventory-10 (BFI-10) [21]
	Mental Health	SF-36 Mental Health Summary Score [22,23]
Physical Parameters	Pain	Pain Rating from 0 to 10 “on the average” from Health and Retirement Study [24]
	BMI	Body Mass Index [25]
	Body Consciousness	Frequency Thinking About One’s Body [26]
	Independent Activities	Functional Activities Questionnaire [27]
	Self-Rating of Health	Vulnerable Elders Survey (VES) [28]
Behaviors	Physical Activity Index	3 Items from Health and Retirement study [24]
	Medication Adherence	Modified Medication Discrepancy Tool (MDT^2^) [29]

**Table 2 geriatrics-07-00111-t002:** Sample Characteristics by Ethnic Group (n = 601).

	African American (n = 115)		European American (n = 217)		Hispanic (n = 127)		Afro-Caribbean (n = 142)			
	M	SD	M	SD	M	SD	M	SD	F	*p*
Age	71.5	7.2	76.7	8.9	73.1	7.7	73.5	7.7	14.03	<0.001 ***
Education	12.5	3.7	15.5	3.8	10.8	5.3	10.9	4.8	49.28	<0.001 ***
Years in US	71.3	7.2	72.6	15.3	34	21.5	40.7	22	221.86	<0.001 ***
Cognition	26.3	3.2	28	2.7	25.6	3.1	25.6	3.8	23.22	<0.001 ***
Function	1.6	3.2	1	2.2	2.4	4.3	2.3	4.3	6.55	<0.001 ***
BMI	33.4	7.4	28	6.3	27.9	4.5	29.6	5.8	21.11	<0.001 ***
Activity Index MET-h/week	4.1	2.9	4.8	2.8	3.6	2.4	4.7	3.3	5.1	0.002 **
									χ^2^	*p*
Sex	21 (M)	94 (F)	85 (M)	132 (F)	25 (M)	102 (F)	35 (M)	107 (F)	21.15	<0.001 ***
Born in US	6 (No)	109 (Yes)	33 (No)	184 (Yes)	8 (No)	119 (Yes)	23 (No)	119 (Yes)	357.5	<0.001 ***
Medicaid	85 (No)	29 (Yes)	198 (No)	11 (Yes)	74 (No)	53 (Yes)	93 (No)	44 (Yes)	71.3	<0.001 ***
Activity Level	48 (L) 40 (M) 27 (H)		63 (L) 91 (M) 63 (H)		51 (L) 60 (M) 16 (H)		58 (L) 31 (M) 53 (H)		35.89	<0.001 ***

Note. Activity Level: L = low, M = medium, and H = high, *** *p* ≤ 0.001, ** *p* ≤ 0.01.

**Table 3 geriatrics-07-00111-t003:** Differences by Physical Activity Group.

	Low Activity		Moderate Activity		High Activity			
	M	SD	M	SD	M	SD	F	*p*
Demographic								
Age	76.2	8.9	74.6	8.5	72.7	7.5	11.08	0.001 ***
Education Years	12.8	4.5	13.7	4.6	14.2	4.3	5.27	0.022 *
Years in US	58.7	24.4	61.9	21.5	55	22.5	0.89	0.346
Cognitive								
Cognition (MMSE)	26.6	3.3	27.3	2.8	27.8	2.7	9.76	0.002 **
Psychosocial								
Social Engagement	4	1.9	5	1.8	6.1	1.8	56	<0.001 ***
Social Network	16.4	5.6	17.8	5.5	19.1	5.5	12.41	<0.001 ***
Spirituality	28.1	9.5	26.9	9.2	26.1	9.9	1.19	0.277
Resilience	78.8	14.3	82.7	12.3	85	10.2	14.39	<0.001 ***
Depression	11	10.4	7.2	7.8	6.6	6.8	16.5	<0.001 ***
Anxiety	30.5	10.3	27.1	7.9	26.5	6.7	9.16	0.003 **
SF36 Mental Health	53.4	11.1	56.3	8.6	56	7.7	5.11	0.024 *
Big 5 Personality								
Anxiety	7.79	1.713	8.27	1.781	7.93	1.866	2.839	0.060
Conscientiousness	8.29	1.729	8.69	1.71	8.55	1.634	2.061	0.129
Extroversion	6.48	2.069	6.48	1.882	6.99	2.121	2.165	0.116
Neuroticism	4.42	2.274	3.96	2.065	4.13	2.016	1.705	0.183
Openness	7.09	2.095	7.65	2.147	7.92	1.881	4.863	0.008 **
Physical								
Pain	3.2	3.6	2.4	3.3	2	3.1	8.47	0.004 **
BMI	31.5	8.8	28.6	5.6	28.2	4.8	15.34	<0.001 ***
Body Consciousness	4.2	1.5	4.1	1.5	4.1	1.5	0.71	0.402
Self-Health Rating	1.0	1.3	1.1	1.6	1.5	1.9	6.52	0.011 *
Functional Activities	3.1	4.8	1.1	2.2	0.8	2.3	32.15	<0.001 ***
Behavioral								
Medication Adherence	3.6	0.9	3.8	0.6	3.6	1	0	0.974

Note. * *p* ≤ 0.05, ** *p* ≤ 0.01, *** *p* ≤ 0.001 [N = 601].

**Table 4 geriatrics-07-00111-t004:** Generalized Linear Regression Results of Predictors selected from the Elastic Net Regression: OLS and EN Predicting average METs per week.

	OLS			Elastic Net			
	β	SE	*p*	β	SE	F	*p*
**Age**	**0.14**	**1.18**	**0.239**	**0.10**	**0.04**	**5.42**	**0.020**
Years in US	−0.05	−0.29	0.774	0.00	0.00		
**Years of Education**	**−0.24**	**−2.62**	**0.010**	**−0.11**	**0.04**	**7.36**	**0.007**
Social Support	−0.04	−0.49	0.625	0.00	0.00		
**Social Network**	**−0.18**	**−2.16**	**0.033**	**−0.11**	**0.04**	**7.46**	**0.007**
Med Compliance	−0.11	−1.39	0.168	0.00	0.02		
Spirituality	−0.29	−3.18	0.002	0.00	0.00		
Resilience	0.08	0.73	0.468	0.00	0.01		
**Depression**	**−0.15**	**1.39**	**0.167**	**−0.14**	**0.04**	**10.52**	**0.001**
Anxiety	−0.05	0.46	0.648	0.00	0.00		
SF-36 Mental Health	−0.09	0.06	0.323	0.00	0.01		
Cognition (MMSE)	0.16	1.70	0.091	0.00	0.02		
**Pain (0–10)**	**−0.07**	**−0.78**	**0.440**	**−0.09**	**0.04**	**5.7**	**0.000**
BMI	0.06	0.71	0.482	0.06	0.04	2.36	0.125
Body Image	0.01	0.07	0.946	0.00	0.00		
Self-Rating of Health	0.05	0.54	0.589	0.00	0.02		
European American				0.04			
African American	0.01	0.08	0.938	0.00			
Hispanic American	−29	1.23	0.043	−0.06			
Afro Caribbean	0.11	1.08	0.282	0.00	0.01		
Born US	0.14	0.92	0.360	0.00	0.02	0.01	0.992
Big Five Personality							
Anxiety	−0.05	−0.66	0.510	0.01	0.03	0.24	0.624
Conscientiousness	0.07	0.76	0.446	0.05	0.03	2.13	0.145
Extroversion	−0.06	−0.70	0.488	−0.03	0.03	0.62	0.431
Neuroticism	0.17	1.83	0.069	0.00			
Openness	0.02	0.21	0.831	0.03	0.03	0.92	0.338

Note. Bolded variables were retained in the final EN selection. The EN used 100 bootstrapped samples. Df = 1 for all of the retained variables except for pain (df = 2). R^2^ = 0.221 with an apparent prediction error = 0.859, expected prediction error = 0.899 and Overall RMSE = 0.081.

**Table 5 geriatrics-07-00111-t005:** Correlation of Retained Predictors and METs Per Week.

Predictors	1	2	3	4	5	6
(1) Hours Per Week						
(2) Age	−0.110 **					
(3) Years of Education	0.087 *	−0.099 *				
(4) Social Network	0.109 *	−0.079	0.117 **			
(5) Depression	−0.186 **	0.011	−0.189 **	−0.198 **		
(6) Pain	−0.143 **	0.027	−0.227 **	−0.046	0.165 **	

Note. * *p* ≤ 0.05, ** *p* ≤ 0.01.

**Table 6 geriatrics-07-00111-t006:** Ordinary Least Squares Coefficients Table of Variables Retained by the EN.

						95% CI	
Predictors	B	SE	β	t	*p*	Lower	Upper
(Constant)	176.55	40.60		4.35	<0.001	96.75	256.34
Age	−0.67	0.46	−0.07	−1.45	0.147	−1.58	0.24
Years of Education	0.20	0.90	0.01	0.22	0.826	−1.56	1.96
Social Network	0.94	0.74	0.06	1.28	0.203	−0.51	2.39
Depression	−1.52	0.43	−0.17	−3.51	<0.001	−2.38	−0.67
Pain	−0.69	1.27	−0.03	−0.54	0.587	−3.20	1.81

## Data Availability

The data presented in this study are available on request from the corresponding author. The data are not publicly available due to privacy concerns.

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
