# Peer review of "Factors Associated with Physical Activity in a Diverse Older Population"

_geriatrics, 2022, doi:10.3390/geriatrics7050111_

Round 1

Reviewer 1 Report

Major comment:

As the authors noted, plenty of papers have studied the potential factors associated with physical activity among older adults. In the introduction, the authors stated that this paper is novel because they examine factors related to physical activity in an ethnically diverse group of older adults. In comparison, prior studies either didn’t study or discuss race/ethnicity information. However, in the discussion, the author did not differ much about how looking at the diverse race group adds to the current evidence base. Using a diverse race group seems to be only for writing a novel paper. Also, some identified factors have been well established in previous studies (e.g., age, education, and depression). The cross-sectional nature of this study did not add much to what has been known. Without further discussion on the implication of using this diverse group, this study seems to lack novelty. 

Minor points:

Introduction

  • “Prior studies evaluating factors associated with physical activity among older adults have reported conflicting results and did not account for collinearity among variables.” Can the authors cite the prior studies here?

Table 1

  • It is unclear what would be the implication of Body consciousness (description: Frequency Thinking About One’s Body) or why this variable is needed.

Author Response

Response to Reviewer 1

            We would like to thank the reviewer for the thoughtful comments and questions. A point by point response follows.

Major Points

  • What is novel about this paper?

We agree that many papers have been published on the factors related to physical activity in older adults. In our review of those papers, however, we found that many were focused on only a single ethnic group or gender (men only or women only), included middle age adults, or used a more limited number of potential predicting factors in their analysis. We  explained this in the following paragraph that was added to the introduction:

            “Gomes8 studied physical inactivity among 19,298 older adults across 12 European countries. The authors found that the prevalence of inactivity was 12.5%, varying from 5% in Sweden to 29% in Portugal. The primary factors associated with inactivity were older age, depression, physical impairments, low meaning in life, low social support, and memory loss. However, the authors did not investigate differences among ethnic groups. A recently published study by AlFaris9 evaluated physical activity levels and factors associated with activity and inactivity in a multi-ethnic population of 1,800 middle-aged men in Saudi Arabia. The authors found that the lowest prevalence of physical activity was 16% in those from the Philippines and the highest was 58% in those from Saudi Arabia, suggesting immigrants may be more inactive. Other factors associated with inactivity were family living, educational level, income, and body mass index (BMI). However, these authors did not investigate the effects of different factors among the ethnic groups, and they did not study older adults. Another study evaluated if social isolation, ethnicity, and gender were associated with inactivity in 3,298 middle and older age adults living in Northern Manhattan10. The prevalence of inactivity was 40.5%. Inactivity was more prevalent among Hispanics (OR: 2.18), in women (OR: 1.33), in those on Medicaid or uninsured (OR: 1.2), and in those who had less than 3 friends (OR: 1.41). This study focused only on social factors controlling for co-morbidities. Additional evaluation of the factors that affect physical activity levels in older adults is still needed.”

As suggested by the reviewer, we cited more studies here and in the Discussion. We also expanded the discussion of previous studies in the Discussion:

            “Other published studies conducted in older adults have targeted population groups in countries such as Ireland22, Jamaica23, Scotland25, India26, Australia27, and Malaysia28 among others. There is a scarcity of research directed at studying differences in factors associated with physical activity within groups (including men and women) based on race or ethnicity. The present study helped to fill this gap, especially with the inclusion of Afro-Caribbeans, a group that has rarely been examined independent of African Americans. Interestingly, our study found that while ethnic group membership was a factor based on a one-way analysis of variance (ANOVA), more rigorous analysis across the many potential predictor variables eliminated ethnic group membership and identified five factors – age, fewer years of education, smaller social network, higher depression, and higher pain – as major contributors to lower participation in physical activity among ethnically diverse older adults. Social network, depression, and pain can be ameliorated by physical activity. They are modifiable factors that can be addressed by physical activity itself. The effects of limited years of education are also modifiable, but this is more difficult, and cannot be addressed by physical activity itself.”

Our argument for the contributions of the paper is based on the analytic approach as well:

“The purpose of this secondary analysis was to examine factors associated with physical activity in a diverse older population. Unique to this study was the use of rigorous analytical techniques to build the most parsimonious, best fitting model predicting older adults’ levels of physical activity. This is particularly important in instances of a large number of variables with high multicollinearity. This study used elastic net regression analysis combining the penalties for both ridge regression and LASSO to investigate the underlying factors that predict physical activity in older adults. For example, instead of just studying ethnic differences in physical activity, we went further and identified the key significant predictors, many of which were highly correlated with ethnicity.

Additionally, when controlling for variability accounted by these underlying factors, the unique variance accounted for by ethnicity alone becomes nonsignificant. The elastic net regression technique is less affected by multicollinearity between these underlying components and ethnicity and therefore allowed us to get a clearer view of these critical underlying components.”

And finally, we summarized the contribution of this paper to our body of knowledge and understanding of the factors that predict activity in the older population.

“While a number of published studies have addressed the question of factors influencing older adults’ physical activity levels, none have employed the large range of instruments/tools, used this robust statistical analysis, and included older adults from multiple ethnic groups as done in this study.”

Respectfully, we believe that it is the rigorous analytic strategy employed that is of equal if not greater value than the inclusion of individuals from multiple ethnic groups as multicollinearity across a large number of factors found to affect levels of physical activity has been a major challenge to this research.

  • Need further discussion of the implications of this study.

Our discussion of the implications of the study results focused primarily on the factors found to contribute unique variance, particularly those that are potentially modifiable, pain, depression, and social network. This is woven into much of the discussion and wrapped up in the following paragraph:

“A complex group of interacting factors that influences participation in physical activity among older adults clearly exists. While these factors may be challenging to overcome, involvement in physical activity can provide enormous benefits including long-term reduction in pain and symptoms of depression and improved social connectiveness among others. Partnerships among local senior centers, low income housing developments, places of worship, YMCAs, and healthcare providers are crucial in the development of tailored multi-faceted programs for physically inactive older adults especially those experiencing pain and/or depression. These programs can provide health-related education pertinent to the identified medical issues (e.g. pain, depression) and assist participants in meeting other participants and in developing specific physical activity-related goals which are known to be associated with sustained involvement.12

Minor Points

  • Cite prior studies reporting conflicting results and not accounting for collinearity among variables

Additional citations have been added to both the Introduction and the Discussion exemplified by the first and second examples under Major Points. Altogether, we added 7 citations.

  • Reasons for including body consciousness in the variables analyzed.

There have been a number of reports that the ideal body image in some ethnic groups favor a larger woman, including several of the ethnic groups in our sample. We added this information and a citation to Table 1. We did not discuss the implications, however, because this variable was found to be insignificant.

  • Methods and Results sections could be improved.

In the Methods section, we added additional detail on the development of the HARI sample and database. In the Results section, some editing was done to improve the flow and clarity of the presentation of the results of the analytical strategy.

Reviewer 2 Report

Dear authors,

Congratulations on the completion of this research work, for your time and dedication.

My comments are very positive about your research.

I congratulate you on the conceptualisation of the problem, the design and method, as well as the discussion of the debate and conclusion. Very elaborate.

I also congratulate you for discussing in depth and showing a broad conceptualisation, as well as showing in detail the method followed. In the introduction you present the benefits of physical activity in healthy ageing and conceptualise the introduction well.

Describe the methods and procedures followed. The strength is the African American, Afro-Caribbean, European American and Hispanic American, diverse ethnic groups in this study. Bringing an important strength to the scientific field is that pain and depression can reduce physical activity, but activity can also help reduce pain and depression.

Here are some suggestions for improvement, in order to improve your citations, downloads, visits, readings from other scientists etc.

-I will clarify or better explain the range of people in the study. It indicates from 56 to 96 years old, from what age do you consider an older person, from 60 years old onwards?

a) what theoretical implications does this work have for the scientists who read this work, for theorists in the field or colleagues?

b) strengths of your work with respect to other studies.

c) future lines of research arising from this work that need to be pursued.

d) expand on citations. There is a recent study done in Spanish elderly people, maybe it is worth to update your bibliography and internationalize your study at Hispanic level in the introduction or discussion of 2022 says: 

1. "Those with high physical activity had better levels of functional ability and autonomy. In addition, dissatisfaction with one's own health is associated with low levels of physical activity. The practice of light physical exercise in older women promotes greater autonomy and functional capacity for activities of daily living, which results in independence in daily life as well as fostering social bonds, as well as obtaining greater satisfaction with their own health, with the socioemotional benefits that this can bring". doi: 10.3390/ijerph18136926

2. "In their study, regular physical activity was associated with higher levels of some of the socio-environmental aspects of quality of life, such as higher levels of education and income, generating social inequalities and modulating lifestyles in older people". DOI: 10.3390/ijerph182010815

I hope it will get better visibility this way! Congratulations, I loved your work!

My sincere congratulations for the work.

Author Response

Response to Reviewer 2

We would like to thank the reviewer for the thoughtful comments and suggestions. A point by point response follows.

  • Clarify the age range from 59 to 96.

Our eligibility criterion for age was 60 and above. However, one participant was just a few days younger than 60 but did not want to postpone the interviews and assessments. For the sake of accuracy, we have reported the range as 59 to 96.

  • Theoretical implications of this work focused upon the factors that contributed unique variance to prediction of levels of physical activity, a discussion of the complexity of these interacting factors and of the bidirectionality of their influence.

  • Strengths of the work in respect to other studies.

We have identified three major strengths:

  1. The multi-ethnic (diverse) character of our sample.
  2. The large number of variables that we were able to test (with the exception of motivation and self-efficacy).
  3. The rigor of the analysis that better addresses the issues of multicollinearity across variables that that have made it difficult to identify the factors that contribute unique variance to the prediction model.

This is summarized in the following paragraph:

“While a number of published studies have addressed the question of factors influencing older adults’ physical activity levels, none have employed the large range of instruments/tools, used this robust statistical analysis, and included older adults from multiple ethnic groups as done in this study.”

  • Future lines of research are found in the Discussion section:

“Both the results of this study and the limitations of this work suggest direction for future research. The bidirectionality of the interactions between some of the variables of interest and need to know, for example, in what instances pain or depression precedes/leads to inactivity versus resulting from inactivity, calls for examination of longitudinal data to understand these relationships. The high degree of interaction among many of the factors affecting physical activity suggest that future studies take a multifactorial, holistic approach encompassing all of the known and even suggested predictors to produce a comprehensive model for physical activity in older adults.

Four of the five significant predictors of physical activity in the older adults studied are at least partially modifiable, calling for the testing of interventions addressing these factors’ potential effect on physical activity and inactivity. A few suggestions include education regarding the effect of activity on common sources of pain such as arthritis or back pain; encouraging providers to write a “prescription” for a daily walk or workout for those with depression, community outreach to isolated older adults, improving the walkability of neighborhoods, repairing sidewalks, adding trails, and making these areas safe to walk and work out.”

  • Expand on citations.

We have done this in both the Introduction and Discussion, adding seven citations. The paragraph added to the introduction exemplifies this:

“Gomes8 studied physical inactivity among 19,298 older adults across 12 European countries. The authors found that the prevalence of inactivity was 12.5%, varying from 5% in Sweden to 29% in Portugal. The primary factors associated with inactivity were older age, depression, physical impairments, low meaning in life, low social support, and memory loss. However, the authors did not investigate differences among ethnic groups. A recently published study by AlFaris9 evaluated physical activity levels and factors associated with activity and inactivity in a multi-ethnic population of 1,800 middle-aged men in Saudi Arabia. The authors found that the lowest prevalence of physical activity was 16% in those from the Philippines and the highest was 58% in those from Saudi Arabia, suggesting immigrants may be more inactive. Other factors associated with inactivity were family living, educational level, income, and body mass index (BMI). However, these authors did not investigate the effects of different factors among the ethnic groups, and they did not study older adults. Another study evaluated if social isolation, ethnicity, and gender were associated with inactivity in 3,298 middle and older age adults living in Northern Manhattan10. The prevalence of inactivity was 40.5%. Inactivity was more prevalent among Hispanics (OR: 2.18), in women (OR: 1.33), in those on Medicaid or uninsured (OR: 1.2), and in those who had less than 3 friends (OR: 1.41). This study focused only on social factors controlling for co-morbidities. Additional evaluation of the factors that affect physical activity levels in older adults is still needed.”

  • Suggestions for additional citations.

We thank reviewer for these suggestions and have added two of them to our paper.

Round 2

Reviewer 1 Report

I thank the authors for their response to my comments. I believe the paper has been much improved and is suitable for publication.